# Situation analysis for delivering integrated comprehensive sexual and reproductive health services in humanitarian crisis condition for Rohingya refugees in Cox's Bazar, Bangladesh: protocol for a mixed-method study

Rushdia Ahmed,[1] Nadia Farnaz,[1] Bachera Aktar,[1] Raafat Hassan,[1] Sharid Bin Shafique,[1] Pushpita Ray,[1] Abdul Awal,[1] Atiya Rahman,[1] Veronique Urbaniak,[2] Loulou Hassan Kobeissi,[2] Jeffries Rosie,[3] Lale Say,[2] Md Tanvir Hasan,[1] Zahidul Quayyum,[1] Sabina Faiz Rashid[1]

For numbered affiliations see end of article.

**Correspondence to**
Ms. Rushdia Ahmed;
ahmed.rushdia@yahoo.com

## ABSTRACT

**Introduction** Rohingya diaspora are one of the most vulnerable groups seeking refuge in camps of Cox's Bazar, Bangladesh, arising an acute humanitarian crisis. More than half of the Rohingya refugees are women and adolescent girls requiring quality sexual and reproductive health (SRH) services. Minimum initial service package of SRH are being rendered in the refugee camps; however, WHO is aiming to provide integrated comprehensive SRH services to meet the unmet needs of this most vulnerable group. For sustainable and successful implementation of such comprehensive SRH service packages, a critical first step is to undertake a situation analysis and understand the current dimensions and capture the lessons learnt on their SRH-specific needs and implementation challenges. This situation analysis is pertinent in current humanitarian condition and will provide an overview of the needs, availability and delivery of SRH services for adolescent girls and women, barriers in accessing and providing those services in Rohingya refugee camps in Cox's Bazar, Bangladesh, and similar humanitarian contexts.
**Methods and analysis** A concurrent mixed-methods design will be used in this study. A community-based household survey coupled with facility assessments as well as qualitative in-depth interviews, key informant interviews and focus group discussions will be conducted with community people of Rohingya refugee camps and relevant stakeholders providing SRH services to Rohingya population in Cox's Bazar, Bangladesh. Survey data will be analysed using univariate, bivariate and multivariable regression statistics. Descriptive analysis will be done for facility assessment and thematic analysis will be conducted with qualitative data.
**Ethics and dissemination** Ethical approval from Institutional Review Board of BRAC James P Grant School of Public Health (2018-017-IR) has been obtained.

### Strengths and limitations of this study

► This situation analysis is among the first that will provide an overview of Rohingya women and adolescent girls' (aged 12-59 years) sexual and reproductive health (SRH) demands and needs, availability and delivery of SRH services, barriers to service uptake and related challenges in Rohingya refugee camps under an acute humanitarian crisis condition.
► The study will use a concurrent mixed-methods design to assess the current situation and understand the community perspectives and facility readiness to provide different SRH services, related gaps and challenges.
► While designing the study, sampling was done using facility delivery rate available in the existing literature body as no prevalence data on other SRH indicators such as family planning, abortion or menstruation were found.
► A potential limitation foreseen within this study is the unwillingness of certain respondents (both community level and facility level) to disclose sensitive information related to SRH practices, service utilisation and health facility records.

Findings from this research will be disseminated through presentations in local, national and international conferences, workshops, peer-reviewed publications, policy briefs and interactive project report.

## INTRODUCTION

To accomplish the target of the 2030 Agenda and the Sustainable Development Goal 3, ensuring healthy lives and promoting

well-being for all at all-ages,[1] the health needs of refugees and migrants must be corroborated.[2] The world has witnessed a rapid increase in the number of refuges over the past few decades.[3 4] Refugees are defined as people who are displaced from their homes and cross international borders due to complex emergencies and disasters.[5] According to WHO and United Nations High Commissioner for Refugees (UNHCR), globally, a total of 68.5 million people have been forcibly displaced by the end of 2017 due to political turbulence or natural disasters, persecution, conflict and violence or human rights violations.[4 6] Estimates from UNHCR (2018) indicate that an estimated 11.8 million people are internally displaced within their own countries, of which 4.4 million are newly displaced persons. During such humanitarian crises, women, adolescent girls and children comprise more than half of the displaced and refugee population and become the most vulnerable groups needing emergency humanitarian response.[7] Being mostly at their reproductive age, women and adolescent girls require access to basic health, safety and well-being needs as well as service delivery including pregnancy, prenatal care, delivery services, postpartum care, family planning services and other reproductive and sexual health-related services.[7] Limited or no access to quality sexual and reproductive health (SRH) services during emergency and crisis conditions puts women and adolescent girls at higher risk of morbidity and mortality that requires utmost importance in terms of service design, delivery and implementation.

Muslims in Rakhine state of Western Myanmar have been facing severe humanitarian crisis since the 1982 Citizenship Law that took away their Myanmar citizenship and right to self-identify themselves as Rohingyas.[8] Many Rohingya diaspora thus took shelter in neighbouring countries, mostly in Bangladesh due to geographic proximity. Although the Rohingyas have been entering Bangladesh since the 1970s, a large influx happened during 1991–1992.[9] Until August 2017, the number of Rohingya refugees (both registered and unregistered) residing in Cox's Bazar was estimated around 213 000 individuals.[10] An outbreak of violence on Rohingya communities in Myanmar on 25 August 2017 resulted in an influx of more than 700 000 Rohingyas in Cox's Bazar, the southeast coastal district of Bangladesh.[10] This created a grave condition for Bangladesh as a hosting country to immediately respond to the urgent needs of such huge refugee population for food, shelter, clean water, health crises, injuries and traumas with more than half of the population comprising women and adolescent girls.[10 11]

Responding to this massive influx into Cox's Bazar district of Bangladesh has stretched the capacity of the already overburdened local administration and health systems.[11] Even though the Government of Bangladesh (GoB), UN agencies, national and international non-government organisations (NGOs) are attempting to respond promptly to the humanitarian crisis for Rohingyas,[11] resolving the crisis needs more integrated contribution from major global players.[11] Furthermore, implementing comprehensive SRH services poses particular challenges in a refugee population due to their vulnerability and transitions and due to lack of clarity on traditional beliefs and cultural models.[12] Although minimum initial service package of SRH (ie, priority set of lifesaving activities to respond to reproductive health needs at the onset of humanitarian crisis condition) are being rendered by several partner organisations in Rohingya refugee camps of Cox's Bazar,[11] access to essential comprehensive reproductive, maternal and newborn health services remains a major concern due to inconsistencies in the quality of services provided and varying implementation of the established minimum package of health services endorsed by various authorities. Thus, WHO is aiming to deliver integrated comprehensive SRH services to meet the immediate SRH needs of extremely vulnerable Rohingya women and adolescent girls who are in acute humanitarian crises. In order to implement such comprehensive SRH service package, a critical first step is to undertake a situation analysis and understand the current state of affairs, cultural and demographic dimensions and capture lessons learnt that is essential for sustainable and successful implementation.[12]

An initial assessment of the current situation in Rohingya refugee camps is pertinent given the acute humanitarian crises and will provide an overview of the needs, availability and delivery of SRH services for adolescent girls and women aged 12–59 years in Rohingya refugee camps. Such exploration can also shed light on the distinctive SRH health needs of Rohingya women and adolescent girls. To explore the facility readiness in providing different SRH services, the gaps in the resources and skills required to provide the comprehensive care, facility level data can also be explored. Thus, a situation analysis in such humanitarian crisis situation will provide a complete understanding of Rohingya women and adolescent girls' SRH demand and needs and barriers to service uptake. Findings from this study will also advance current understanding of implementers like WHO, Inter-Agency Working Group and other key stakeholder on where and how to tailor and improve management and delivery of comprehensive SRH services. This will also allow to explore the possibility of updating and standardising service and training packages for SRH service providers and drawn on in future to improve delivery and utilisation of comprehensive SRH services in similar humanitarian crises contexts in Bangladesh and other low-income and middle-income countries.

## STUDY OBJECTIVES

Overall objective of this study is to conduct a situation analysis to assess demand and supply side barriers in accessing SRH services by adolescent girls and women aged 12–59 years in Rohingya refugee camps of Cox's Bazar, Bangladesh.

Specific objectives include:
1. To assess SRH needs and service seeking behaviour of Rohingya women and adolescent girls.
2. To conduct assessment of facility readiness and explore availability of resources (manpower and essential kits, drugs and supplies), measure gaps and estimation of cost of resources for providing comprehensive SRH services to Rohingya women and adolescent girls.
3. To explore demand and supply side challenges in seeking and rendering SRH services.
4. To explore scope of improvement of the existent SRHR service delivery system.

## METHODS AND ANALYSIS
### Study design and population
This study will employ a concurrent mixed-methods study design using both qualitative and quantitative techniques. A community-based survey coupled with facility assessments as well as qualitative in-depth interviews (IDIs), key informant interviews (KIIs) and focus group discussions (FGDs) will be conducted with a broad range of stakeholders. The primary study population is adolescent girls and women (12–59 years) who were forcibly displaced from the Rakhine state of Myanmar and migrated to Bangladesh since 25 August 2017 and residing in the refugee camps of two selected subdistricts of Cox's Bazar district. The secondary study groups include Rohingya

males; influential community members; formal and informal healthcare providers; and government, international/national non-government organisations (INGOs/ NNGOs) programme staff. Duration of the study is 1 year starting from 20 July 2018.

### Study site
The study will be conducted in Rohingya refugee camps of Ukhiya and Teknaf subdistricts in Cox's Bazar district in the southeast coast of Bangladesh. In total, 34 refugee camps are located in these sites as per health sector, Cox's Bazar, June 2018 data, where only two are registered and pre-existing settlements. The congested camp environment along with fragile forest, hilly terrain induced geographical difficulty and seasonal variation including monsoon and rainfall, make the life of Rohingya diaspora critical.

### Data collection
The following methods will be applied for data collection (figure 1).

#### Household survey
A household survey will be conducted among the Rohingya refugee adolescent girls and women (12–59 years) to understand their SRH needs, service utilisation and barriers to access and use services. The survey will capture information related to health and care seeking

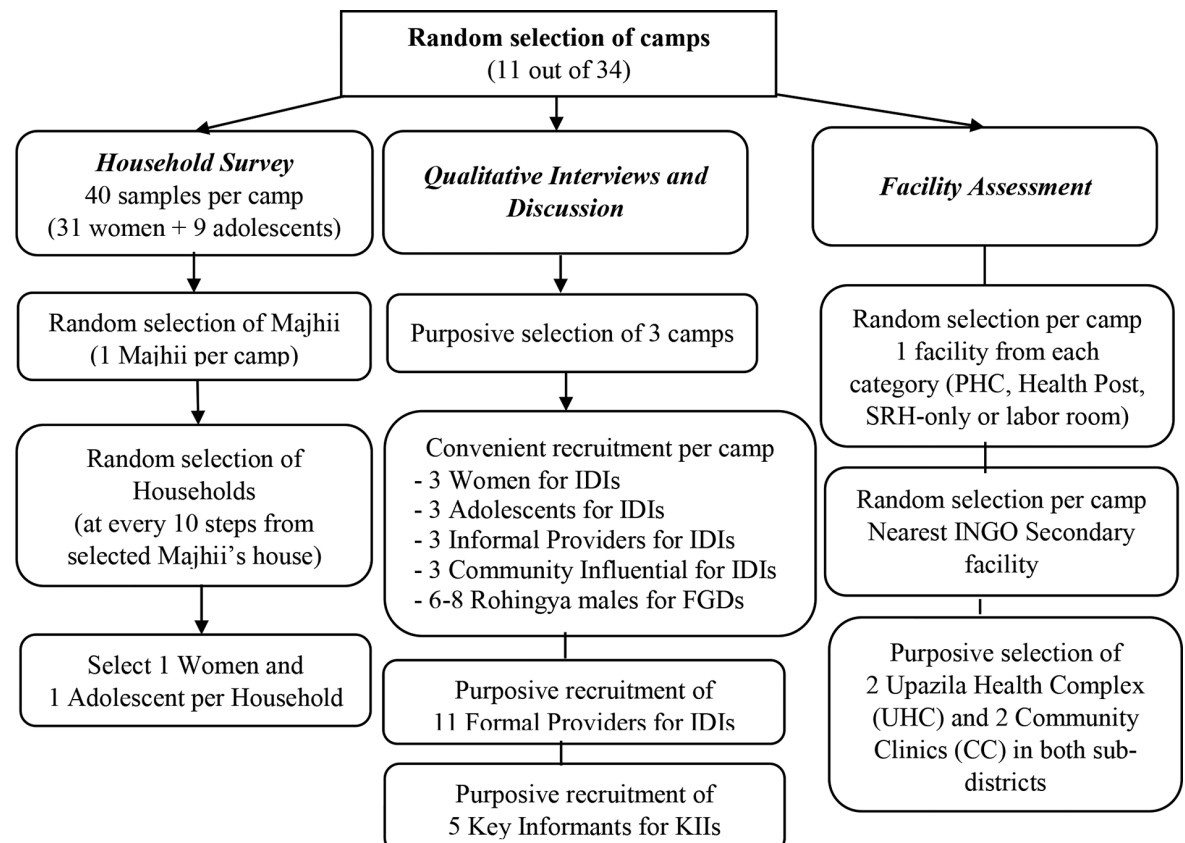

**Figure 1** Sampling techniques to conduct situation analysis. FGDs, focus group discussions; IDIs, in-depth interviews; INGOs, international non-government organisation; KIIs, key informant interviews; PHC, primary healthcare.

of Rohingya women and adolescent girls, especially on menstrual health, pregnancy and delivery care, postnatal care, family planning services used, menstrual regulation (MR) and abortion, sexually transmitted diseases, service utilisation and barriers related to accessing services. This will enable us determine the gaps in availability and utilisation of SRH services.

### Sample size and sampling techniques

Sample size: considering available data sources on Rohingya refugees, age and gender breakdown of total refugees were identified from Bangladesh Refugee Emergency Population Fact Sheet.[13] The total number of Rohingya women and adolescent girls aged 12–59 years is 269 345, where 60 084 are adolescent girls aged 12–17 years and 209 261 are women aged 18–59 years. According to United Nations Population Fund (UNFPA) Monthly Situation Report on Rohingya Humanitarian Response,[14] on May 2018, 22% of pregnant Rohingya women and adolescent girls gave birth in health facilities. Considering this as the prevalence rate with 95% CI, 5% margin of error and 1.5 design effect, the estimated sample size for the household survey is calculated to be 395. Due to large study population (269 345), considering the finite population correction,[15] the estimated sample size is 395. We considered 10% non-response rate and plan to reach 440 women and adolescent girls.

The formula used for sample size calculation:

$$n = \frac{z^2 p(1-p)}{d} \times \text{deff}$$

Among 269 345 study population, 22.3% (60 084) were adolescent girls aged 12–17 years and 77.7% (209 261) were women aged 18–59 years. So, the sample has been distributed proportionally among the two groups (table 1).

Sampling techniques: a multistage sampling technique will be employed for selecting camps and study respondents.

*Stage I:* in the first stage sampling, refugee camps in Ukhiya and Teknaf subdistricts where new makeshifts were established after the 25 August 2017 influx will be selected. Registered refugee camps with old settlements where Rohingya refugees are living since before the influx of 25 August 2017 will be excluded. Eleven camps will be randomly selected from 34 camps, which represents 30% of the total camps in Ukhiya and Teknaf sub-districts at Cox's Bazar. The samples will be equally distributed among the 11 selected camps (40 sample per camp). Our randomly selected camps in the Ukhiya Upazila are Camp-1W, Camp-3, Camp-5, Camp-7, Camp-10, Camp-11 and Camp-17, and in the Teknaf Upazila are Camp- 21, Camp-22 and Camp- 26.

*Stage II:* in the second stage sampling, a complete list of Majhiis (local community leaders of Rohingyas) of the selected 11 camps will be collected from the camp in-charge office and one Majhii will be randomly selected per camp.

*Stage III:* survey data will be collected from 31 women and 9 adolescent girls aged 12–17 years from each camp considering proportionality in these groups. The house of the selected Majhii will be determined as starting point to select the sample households. Selected from both left and right side, the first (closest) sample household will be after 10 footsteps from Majhii's house. Accordingly, households in every 10 footsteps will be selected until desired number of respondents are interviewed. One woman will be picked from every selected household as a respondent. If more than one woman is found in a household, then one will be randomly selected on the basis of availability and interest for interviewing. In addition, if available, an adolescent (aged 12–17 years) will be interviewed from the same household. If more than one adolescent girl is found in that household, then similar procedure will be followed.

### Data collection methods and tools

A structured quantitative questionnaire (online supplementary file 1) will be prepared following a guideline prepared by UNFPA and Save the Children for humanitarian crisis situation[16] and scholarly literatures.[17] Pretesting will be done in similar camps that are not selected for the study before initiating data collection. Data will be collected through Samsung tablets (Model no. SM-T231) by using SurveyCTO software, an Open Data Kit (ODK) tool widely used for collecting survey data. Using ODK will ensure automatic data storage in database that will be converted to statistical software package Stata V.13 (StataCorp, 4905 Lakeway Drive, College Station, Texas, USA) for data cleaning and analysing. The questionnaire will be translated in Bangla from English including key SRHR terms in Rohingya language. Local dialects will be used in the questions for clarity of our study objective-related topics (local language). Local experienced female interviewers, who understand the language of Rohingya

| Table 1 | Distribution of sample size for household survey | | | |
|---|---|---|---|---|
| Primary study population | Total number of individuals | Percentage (%) | Sample size | Sample size considering 10% non-response |
| Adolescent girls aged 12–17 years | 60 084 | 22.3 | 88 | 99 |
| Women aged 18–59 years | 209 261 | 77.7 | 307 | 341 |
| Total number of women and adolescent girls aged 12–59 years | 269 345 | 100 | 395 | 440 |

community, will be recruited for data collection. Extensive training sessions will be conducted to orient and train them regarding the study objective and tools. The interviewers will be monitored by two researchers and one statistician of BRAC James P Grant School of Public Health while collecting data. In addition, two local male interviewers will be recruited for building rapport with the community key persons in each camp.

### Data analysis

Descriptive analyses will be performed on survey data collected to understand sociodemographic characteristics, need for SRH services, healthcare and service seeking behaviour, service utilisation patterns and barriers to accessing services, challenges faced on the basis of distance and waiting time at health facilities, their restrictions and reasons for not taking services and so on. Statistical analysis will be performed in separate groups for women (18–59 years) and adolescent girls (12–17 years) to understand their specific SRH needs. Depending on the distribution of variables, frequencies, percentages, mean (SD) and range as summary statistics will be reported. $\chi^2$ test will be performed to measure the association between sociodemographic characteristics and other variables of menstrual health, pregnancy, delivery care, family planning services, MR, abortion and sexual transmitted diseases and feasible challenges of demand side and pattern of health seeking behaviour for sexual and reproductive healthcare. Multivariable regression analysis will be conducted afterwards if significant associations are found in bivariate analyses.

### Qualitative interviews and group discussions

In order to complement the household survey findings, we will also conduct IDIs with adolescent girls and women. This will help to further understand their perspectives about the SRH services available and the challenges they face in accessing and using those services. IDIs with formal and informal healthcare providers who are working in the selected refugee camps will be done to understand the barriers and challenges in providing SRH services to the Rohingya refugee adolescent girls and women. To understand the perspective of their male counterparts, FGDs with Rohingya males will also be conducted. KIIs with the key stakeholders from government, INGOs and NNGOs will be conducted to get insights about the existing SRH service delivery system and management challenges. Rohingya community leaders such as Majhiis, religious leaders such as Imams and teachers will also be interviewed (KII) to explore their influence on the adolescent girls and women in using SRH services.

### Sample size and sampling techniques

For qualitative interviews, three camps will be selected from 11 camps (where household survey and facility assessment will be conducted) depending on geographic location, challenging terrains, remoteness, difficulty in accessibility, availability of infrastructure and so on. The sampling strategy and type and number of respondents for each of the qualitative activities planned is provided in table 2. Qualitative data collection will be continued until data saturation is achieved.

### Data collection methods and tools

Separate guidelines will be developed for IDIs and KIIs with different groups and FGDs with males. All tools will be finalised after pretesting in similar camps that are not selected for this study. Qualitative interviews will be conducted by an experienced group of researchers trained in qualitative interviewing and analysis. However, due to language barrier, interpreters

| Table 2 | Sampling frame and characteristics for each qualitative activity | | |
|---|---|---|---|
| **Activity and focus** | **Sampling strategy** | **Respondent groups** | **Estimated numbers** |
| Activity 1: KIIs | Opportunistic/emergent sampling Snowball sampling | Local and international NGO programme leads, managers, SRH focal points, influential workers, government high officials and programme managers. | 5 |
| | | Influential community members: Majhii, imam, female imam and Burmese teachers. | 9 |
| Activity 2: IDIs | Purposive sampling | Rohingya women aged 18–59 years. | 9 |
| | | Rohingya adolescent girls (12–17 years old). | 9 |
| | Opportunistic/emergent sampling | Formal providers such as midwife, health centre in-charge, doctors, nurses and community health workers. | 11 |
| | | Informal providers such as traditional birth attendants, Burmese doctor and traditional healers. | 9 |
| Activity 3: FGDs | Purposive sampling | Rohingya adult males. | 3 FGDs with maximum 6 participants. |

FGDs, focus group discussions; KIIs, key informant interviews; NGOs, non-government organisation; SRH, sexual and reproductive health.

will be recruited from the nearby locality who understand the language and dialect of Rohingya community. Training sessions will be conducted to orient and train them on study objective and qualitative tools prior interviewing. A period of rapport building with the community key persons in each camp site will be critical to the success of this research given known difficulties in accessing the Rohingya population, their conservative cultures, suspicion about motives and postinterview repercussions. Interviews with women and adolescent girls will only be conducted by female researchers and interpreters due to conservative nature of the local population and nature of the questions involved. These dynamics must be handled carefully or else access will be hampered. Networks with influential and key locals will be important in opening doors and initiating discussion. Male researchers including local male interpreters will conduct FGDs, IDIs and KIIs with community males.

### Data analysis

An outline of the plan for qualitative data analysis will be prepared in advance of the research, which will include defining a priori codes according to study objectives. All interviews will be recorded provided consent has been obtained, along with simultaneous note-taking in case of equipment failure. Data transcription will occur immediately following each interview, followed by translation. Data familiarisation will involve reading transcripts repeatedly to surface emerging themes, assess strengths and weaknesses of interview techniques and identify any missed opportunities for further exploration. Transcripts will be reviewed carefully, and coding will be done following the a priori code list. A team approach to analysis will be employed to minimise individual biases. Intracoder and intercoder reliability will be checked. This approach is applied in all aspects of analysis including coding, with multiple analysts coding the same sections of text to assess intercoder reliability. Emerging themes and patterns in the data will be tested using data displays that allow more systematic analysis of the qualitative data. Any emerging codes identified during analysis will be added in the code list after confirmation as a team and will be used for coding all transcripts.

### Facility assessment

A facility assessment will be undertaken to get an overview about the supply side barriers in terms of infrastructures, human resources including training needs, provision and utilisation of SRH services and medical supplies for serving Rohingya refugee population. This facility assessment exercise will help assess facility readiness to provide comprehensive SRH services. An estimation of resources required for providing comprehensive SRH services in the camps will also be done with the data from facility assessment and secondary sources.

### Sample size and sampling techniques

Five categories of health facilities will be chosen according to WHO Health Facility Register (shared internally by WHO). The categories include primary health centres (PHCs), health posts (fixed and plus), labour rooms or SRH only facilities, secondary health facility and community clinics. The first three types of facilities (PHCs, health posts plus and fixed health post and labour rooms/SRH only facilities/maternity centres) are camp specific and situated inside the camp. The two other types of facilities (secondary hospitals and community clinics) are situated outside the camps. One facility from each category will be randomly selected for assessment in each camp depending on the availability. In doing so, health facility listing of the selected camps for this study will be carried out from the WHO Health Facility Register. In the camps, if one facility is available in a selected category, then that facility will be assessed. For example: if there is only one PHC in a specific camp, then assessment of that particular facility will be conducted. In some cases, there are more than one facility of a single category available inside one camp (such as three PHCs in one camp). In such situation, one facility will be selected for this study using random selection method.

With this procedure, the following facilities in table 3 will be assessed in the Ukhiya and Teknaf upazila.

For secondary health facilities, in both subdistricts, two government secondary facilities (Upazila Health Complex) are serving as the main referral points along with other secondary facilities. Additionally, two community clinics are GoB-run primary level facilities under Ministry of Health and Family Welfare. All these health facilities will be selected for assessment. Therefore, we will be conducting facility assessments in total 29 health facilities—11 health posts (fixed), 9 PHCs, 2 labour rooms and SRH only facilities, 5 secondary health facilities and 2 community clinics altogether (table 3).

### Data collection methods and tools

A structured English questionnaire (online supplementary file 2) will be prepared for facility assessment following WHO Service Availability and Readiness Assessment tool.[18] After pretesting and finalising, two researchers will collect data from different categories of health facilities identified. Data will be collected through Samsung tablets (Model no. SM-T231) by using KOBO software, an ODK tool.

### Data analysis

Descriptive analysis will be conducted according to the type of facilities to understand the facility readiness and challenges faced by supply side. The analysis will be performed separately for five different categories proposed to identify gaps at all levels: service provision and availability, service utilisation, human resources including their training, infrastructure and supply of equipment and drugs for providing SRH services by the health facilities.

**Table 3** Types and numbers of facilities for facility assessment

| Upazila | Camp name/ number | Categories of health facilities | | | | |
| | | Health post | Primary healthcare facility | Labour room or SRH only | Secondary health facility | Community clinic |
|---|---|---|---|---|---|---|
| Ukhiya | Camp-1W | 1 | – | 1 | 4 | 1 |
| | Camp-3 | 1 | 1 | 1 | | |
| | Camp-5 | 1 | 1 | – | | |
| | Camp-7 | 1 | 1 | – | | |
| | Camp-10 | 1 | 1 | – | | |
| | Camp-1 | 1 | 1 | – | | |
| | Camp-17 | 1 | 1 | – | | |
| | Kutupalong RC | 1 | 1 | – | | |
| Teknaf | Camp-21 | 1 | 1 | – | 1 | 1 |
| | Camp-22 | 1 | 1 | – | | |
| | Camp-26 | 1 | – | – | | |
| Subtotal | | 11 | 9 | 2 | 5 | 2 |

SRH, sexual and reproductive health.

### Data triangulation
This concurrent mixed-methods study aims for results point of integration as identified by Schoonenboom and Johnson.[19] Data collected from multiple sources such as household survey, facility assessment, qualitative interviews (IDIs & KIIs) and discussions (FGDs) will be triangulated to understand the overall SRH needs, demands, challenges, barriers to access and service provision to the Rohingya refugee women and adolescent girls aged 12–59 years. After initial descriptive analysis of each qualitative and quantitative component, integration of different components to link and explain different dimensions such as SRH needs, SRH service utilisation, barriers to access SRH service and so on will be made. A joint display of qualitative and quantitative findings will also be adopted.

### Patient and public involvement
The research questions and outcome measures of this study were chosen based on the priority of comprehensive information need on the Rohingya diaspora in a complex humanitarian condition in discussion with relevant stakeholders including possible comprehensive SRH service implementers and policymakers. However, neither patients nor public were involved in developing this study protocol.

### ETHICS AND DISSEMINATION
This study poses no more than minimal risk to subjects. Respondents will be asked for written consent prior interviewing. Written assents will be taken from adolescent respondents, and written consents will be sought from their guardians. In case of low literacy of respondents, verbal assent and consent will be sought.

Tape recorders will be used for recording the qualitative interviews in order to collect full and intact thoughts.

Strong password-protected server or user profile will be created and used for quantitative data collection using ODK tools—SurveyCTO for survey and KoBo Toolbox for facility assessment. All forms of data related to this study will be stored in locked storage or controlled-access folders allowing access by authorised persons related to the study, that is, principal investigator, other study investigators and Institutional Review Board (IRB) members of BRAC JPGSPH.

Findings from this research will be disseminated at various levels so that evidence generated can be advocated and translated into policy actions for better SRH of Rohingya refugee women and adolescent girls. This study aims to strengthen SRH service provision through incorporating changes recommended from this research findings, for example, on programme implementation, challenges reported by study population and service providers and so on. Policy briefs with recommendations specific to programme implementation and related challenges will be developed and disseminated to all higher level stakeholders. Stakeholder engagement (Ministry of Health and Family Welfare, GoB; Health sector headed by WHO and SRH subsector by UNFPA) since inception of the study will be beneficial in influencing local policy on SRH service provision. These actors are expected to take lead for service provision changes proposed by the situation analysis and implement necessary amendments in local policy strengthening. Findings will also be presented to relevant local administrators, development partners and NGOs and other relevant parties, academicians and researchers through local and national conferences, dissemination workshops, interactive project report and policy briefs. Additionally, scholarly publications in peer-reviewed journals and presentations in international scientific forums, conferences and symposiums will be done for international audiences.

## EXPECTED CHALLENGES

First and foremost, the challenge expected by researchers of this study is the language barrier as Rohingya people cannot speak or understand Bengali or English languages. Hence, it will be difficult to understand their dialect for the researchers unless trained interpreters are involved. Thus, we will recruit data collectors cum interpreters from the local Bangladeshi community who can speak and understand Rohingya language fluently. They will act as key persons to establish communication between researchers and Rohingya people. However, a list of key terms in Rohingya language and translation into Bangla and English will be prepared with the help of data collectors as a reference for the researchers who will be accompanying the local data collectors to combine their critical thinking with the community dialect. They will also perform spot checks while collecting data. Another challenge is the sensitive nature of the questions to be asked in this study given the conservative culture of the Rohingya population. Only female data collectors and researchers will interview female respondents and male data collectors, and researchers will conduct interviews and FGDs with male respondents. Data collectors will be trained to respect the current humanitarian crises condition in the Rohingya camps, the trauma this community has experienced, their culture, choices and privacy of the interview respondents. Another major challenge can be the entry into local Rohingya community. Harnessing the power relations and dynamics in the Rohingya community is the key to conduct such situation analysis. Thus, the study is designed accordingly to draw on the benefits of engaging gatekeepers to create access in the community for data collection. After entering in each camp, the researchers will communicate with the selected Majhiis, the local Rohingya leaders, and explain him or her the purpose of the study. After leveraging sufficient time to build rapport with the Majhiis, their support will be sought to get access to the households. In terms of working in facilities, accessing the records could be difficult, hence, the higher authority of each facility will be informed beforehand. Another challenge in this study can be political unrest like strike for which data collection may delay. Finally, weather and geographical difficulties is one of the biggest challenges in working at Rohingya camps as located in hilly areas and challenging terrains. In order to avoid any sort of accident, adequate logistics support to the data collectors and study researchers will be ensured.

**Author affiliations**
[1]BRAC James P Grant School of Public Health, BRAC University, Dhaka, Bangladesh
[2]Reproductive Health and Research, WHO, Geneva, Switzerland
[3]Health Sector Coordination Office, WHO, Cox's Bazar, Bangladesh

**Contributors** RA, NF, BA, RH, SBS, PR, AA, AR, MTH, ZQ and SFR contributed in conceptualisation or design of the study, and VU, LHK, JR and LS reviewed and incorporated their critical inputs. RH, SBS and PR have contributed equally. Also, VU, LHK, JR and LS made equal contributions. RA drafted the initial version with support from NF and BA. MTH, ZQ, SFR reviewed and helped revise critically. RH, SBS, AA, PR, AR, VU, LHK, JR and LS reviewed critically for important intellectual content. RA finally revised the version submitted with inputs from all other coauthors. All authors finally approved the version published. RA, BA, MTH, ZQ, SFR and LHK are in agreement to be accountable for all aspects of the work in ensuring that questions related to the accuracy or integrity of any part of the work are appropriately investigated and resolved.

**Funding** Funding is received from World Health Organization (WHO) for this work: grant number: 201991701.

**Competing interests** None declared.

**Patient consent for publication** Not required.

**Ethics approval** This has received ethical approval from Institutional Review Board (IRB) of BRAC James P Grant School of Public Health (2018-017-IR).

**Provenance and peer review** Not commissioned; externally peer reviewed.

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
