## [Reviewer comments · BMJ Open]

ARTICLE DETAILS

TITLE (PROVISIONAL)	Situation analysis for delivering integrated comprehensive sexual and reproductive health services in humanitarian crisis condition for Rohingya refugees in Cox's Bazar, Bangladesh: protocol for a mixed-method study
AUTHORS	AHMED, RUSHDIA; Farnaz, Nadia; Aktar, Bachera; Hassan, Raafat; Shafique, Sharid; Ray, Pushpita; Awal, Abdul; Rahman, Atiya; Urbaniak, Veronique; Kobeissi, Loulou; Rosie, Jeffries; Say, Lale; Hasan, Md Tanvir; Quayyum, Zahidul; Rashid, Sabina Faiz

VERSION 1 - REVIEW

REVIEWER	Professor Sarah Larkins Associate Dean, Research, College of Medicine and Dentistry James Cook University Australia
REVIEW RETURNED	16-Dec-2018

GENERAL COMMENTS	Thanks for the opportunity to review this interesting protocol paper for a mixed methods study about SRH health in Cox's Bazaar. It addresses a critical, high priority area in a vulnerable population, and on the whole is well justified, clearly written and well thought out. The background makes the case for the importance of strengthening SRH services for this population and the study objectives are clear overall. STROBe guidelines are clearly followed for the study protocol, and the various elements of the mixed methods study are well described. Sampling strategies for the different components, especially the household survey, whilst complex are clearly written. Ethical issues are considered and some of the cultural and language issues that will be important to overcome for successful conduct of the study are well handled. I have several suggestions for consideration by the authors that I think may strengthen the protocol further. 1. Overall, I would recommend embedding the whole study protocol in a implementation research approach as recommended by the WHO (https://www.who.int/alliance-hpsr/resources/implementationresearchguide/en/). The main addition to what is proposed would be to increase stakeholder engagement from the early stages of the project and to take a more explicit approach to recommendations for action from the findings - this is currently implied but not crystal clear. Even if the authors decide not to add an IR approach, I think more
---

	information in terms of dissemination with the aim of strengthening service provision and influencing local policy is important. 2. The study is described as using concurrent mixed methods. Whilst this is fine as an approach, and the purpose is clearly complementarity and triangulation of results, the data analysis and interpretation section has no information about how the findings from the data collection methods will be integrated and interpreted jointly. So, either describe it a a multimethods study design within an IR framework, or if mixed methods, need to describe data integration as part of analysis. 3. I found it a bit confusing jumping between the three data collection methods in each of the sampling, data collection and data analysis sections. I wondered whether it would be clearer to introduce the three data collection methods, and then describe each in total in turn. That is - household survey - sampling strategy, data collection tool, data collection and data analysis. Qualitative interviews - sampling strategy, data collection, data analysis, Facility survey - etc etc. 4. In terms of qualitative analysis - it sounds like the authors are describing a kind of abductive analysis as described by Tavory and Timmermans (2014) - ie. initially deductive coding using a priori codes, but then allowing new codes to arise and adding them. 5. Small thing, but in the ethics section, should be assent rather than ascent for adolescent girls.
--	---

REVIEWER	Brynne Gilmore
	Centre for Global Health, Trinity College Dublin Ireland
REVIEW RETURNED	22-Jan-2019

GENERAL COMMENTS	Dear Authors: It is wonderful to see such an important and valuable study being considered, and great too that you are wanting to share your plans via this protocol. I have provided below some topics/points that need further clarification, and some concerns I have on some aspects of protocol. While I think the study itself has been well considered, I feel the protocol can be improved to better relay information to the readers and provide more clarity on the study overall. Towards the end of this review, I have provided some suggestions that might help strengthen your study design. Major:  1. The limitation listed in Abstract is not a 'methodological' limitation per se. I would rephrase to just, "a potential limitation foreseen within this study is the unwillingness...". 2. It would be helpful to explain the difference between comprehensive and basic package of SRH in the literature/background. 3. In the abstract, you note that the focus will be on Family Planning and safe abortion services, but this is not discussed anywhere else within the manuscript. Please note how this focus will be seen within your methodology, or revise accordingly. 4. There is very limited information on the study sites. Please expand to ensure the readers have a more accurate picture of the context.
---

5. Please review for grammar. While the article is well written, at times there are awkward sentences or those that could benefit from addition, or removal, of some punctuation. I have not identified all the sentences within this report.
6. Please explain where the 22% that gave birth in health facilities factored into your sample size calculation (I think you're saying this was used as prevalence rate, but a little unclear), and why you used this as your rate? I thought comprehensive SHR services would go beyond facility? Also, if your primary focus (as stated in intro) is family planning and abortion, wouldn't these estimates be better to use for sample size?
7. Sampling – several issues with this: 1) Please provide more clarity on methods. You are doing multi-stage sampling with both cluster and simple random sampling it appears?; 2) You note (line 6, pg. 9) 'proportionality', but there is no indication that you are doing proportionate sampling. If not, justify why you are not intending to do proportionate sampling as I presume both the camp and community sizes vary; 3) Facility sampling is very confusing –are you doing one of each category (5 categories), in each of your camps? Or in total? I am unclear how you are getting the numbers at the end of this section; 4) I think that all sampling sections section can be simplified (it is very long, and also includes a diagram) and clarified a fair bit. It is too wordy and somewhat confusing.
8. Unclear why including an adolescent from the same Household, might need more explanation and/or note the potential limitations of this (see point below on this specifically)
9. Line 16, pg. 9 – married women, this was not noted in your participation criteria that they had to be married (see note below on study populations).
10. Please include your intended study dates.
11. Ethics – I am unsure if this study poses no more than minimal risk. I can imagine SRH topics (especially FP and abortion), can be quite risky to discuss and may in fact put people at risk if any information was to be released. How will you make sure people are aware of this risk,

and what measures will you put in place to protect them? How will you collect consent for persons with low literacy?
12. Please clarify ethics – have you sought (pg. 12 line 9) or have you been granted (pg. 14). Please provide proof of ethics via certificate (or at least approval number).
13. The synthesis/situational assessment bit of the study is lacking. What is your plan for synthesising your data to provide an overall 'situation analysis'. Do you have any plans for looking at the specific camps data (i.e. are there relations between 'stronger' facilities and the survey respondents), etc.
14. Under data collection methods - please indicate the topics that will be included within the Household survey, as this section has very little about the tool itself, only on the collection method. If you have any of your tools developed (facility assessments or quantitative) it would be good to include as supplementary file.
15. Age-sets: as noted below under study considerations, I am cautious that considering all women age 18-59 as one homogenous group (despite there being a 41 year age range), does not go towards person-centered SRH service design. Please include how you will ensure you are collecting data that can contribute to designing such SHR comprehensive service packages.

Minor:

- Line 7-8 in the abstract – what lessons learned are you trying to capture? From the minimum initial package, for scaling up to the comprehensive? This sentence could use some rewording. Also, what current dimensions? Of the package, of the context...?
- Last sentence of abstract – first line is awkward and needs grammar check. Also, is the situation analysis providing information relevant to similar low-and middle-income contexts, or to similar humanitarian crisis?
- More details on your methods would be helpful in the abstract. What kind of community- based survey? What is the purpose of it? Would help to note within abstract. What types of interviews and discussions? (note KIIs, IDIs and FGDs as per table 2). Who are the stakeholders involved in the survey and the discussions (you note community people, but you have indicated numerous stakeholder groups within the body).
- Point two “this study employs a mixed-method...” under Strengths and Limitations is somewhat awkward, please review grammar.
- Abstract – service package, should this be service packages? Again in 7th line.
- SD Goals to SD Goal, line 6?
- Please review your referencing style. Typically, the full stop goes after the citation (i.e. humanitarian response [7].”
- Line 16-18 – would they be classified as refugees then? According to your definition, no. Maybe change to ‘..of which 4.4 million are newly displaced persons’.
- Line 20 – group to groups.
- Pg. 7, line 19: is this to mean that your inclusion criteria is that they, a) must be forcedly displaced from Rakhine specifically, b) must not have left Myanmar before August 25, 2017. What about women who have entered/experienced menopause, will they be included? What about the inclusion of men (pg. 7 line 43)? Do they have to have a partner/spouse within this age group? What is their inclusion criteria?
- Pg. 8, line 15. Is that the total number all over, or just in your study sites.
- Pg. 8, under Stage I: Would be better to report the inclusion/exclusion of camps first, and then the selection of the camps within the ‘new makeshift camps’. Currently this is confusing, and makes it seem like you select 11, and then you select more from the new ones.
- Table 2 – clarify that you mean 3 FGDs, not 3 participants (like I believe the rest of your columns indicate)
- Line 26 pg. 9 – unsure what is meant by ‘depending on geographic location...’ are you trying to get only remote/difficult places, do you want a balance? Will the KIIs be from these 3 sites, or are KIIs not at this level?
- Pg. 10, line 15 – what do you mean ‘based on proximity random selection will be conducted’?
- Pg. 10, line 17 – you list only one secondary facility, but note there are two.
- Figure 1 (appendix) note that you will collect from 30 and 10 women and adolescents, respectively. This is different than in the body of the paper. Please review diagram.
- Pg. 11, line 44 “Group discussions... tested using data displays...” please explain this?

	 • It could be important to include in your dissemination plans how you intend these findings to influence SRH programming/packages within Cox's Bazar. Essentially – revisit why you are doing the situational analysis in the first place. Study considerations:  • Consider your population groups – I imagine within Rohingya some married women and unmarried women might have different needs, and/or at least need to discuss these in different groups (same goes for men). Might therefore be good to distinguish between either 'adolescent' married or unmarried, and non-adolescent married and unmarried, within your interviews and IDIs (also relevant for pg. 7). This should also be considered for your FGDs with men – I would strongly consider conducting different ones (married, unmarried, adolescent, older etc). It could also be beneficial to try to complement your male FGDs with a couple IDIs with men, as often sensitive topics like family planning and abortion might be difficult to discuss in more open forms. • You may also need to do more than 3 FGDs with men. I would not be surprised if (at least within married couples), men can heavily influence SRH access and demand. However, this goes to a point below where I suggest designing/conducting qualitative after you have analysed surveys. • As noted above, please consider not using adolescents from same households as non- adolescent. First, they will know the types of questions you asked each other, which could entrench on ethical issues. Second, both respondents would likely have similar physical access to services (though obviously other barriers), which might skew some of your results. • It might be helpful to analyse your quantitative data first, and then decide on some of your qualitative work, especially in terms of whom to interview and the study sites. You might find that different communities have different survey responses, depending on their access etc, so to only do 3 sites, when there could be variation in responses across say, 5 sites, would be limiting and not give you full picture. Further, Qualitative if you design your guidelines (and thus analysis framework) after the quantitative has been analysed you could get more rich and purposeful data if you base some of the discussions on what was arising from the survey (i.e. can really focus on the main barriers are reported in survey, and why/how these arise, what can be done etc). This could give you the chance to do a more in-depth exploration of some of the key issues, and also serve to triangulate your previous findings. If this is something you would consider, then need to adjust some of the body to reflect this. • Might be appropriate to stratify your analysis even further- after doing 18-59, could also do like 18-40? Or another age group. With this over 40-year age difference between 18-59, needs etc. are very likely to be different. Person-centred SHR services would need to be responsive to each age-set needs, and not treat like one homogenous group. I think you should also plan to do your interviews with various age-ranges.
--	---

Reviewer: 1

Reviewer Name: Professor Sarah Larkins

Institution and Country: Associate Dean, Research,

College of Medicine and Dentistry

James Cook University

Australia

1. Overall, I would recommend embedding the whole study protocol in a implementation research approach as recommended by the WHO (<https://www.who.int/alliance-hpsr/resources/implementationresearchguide/en/>).

The main addition to what is proposed would be to increase stakeholder engagement from the early stages of the project and to take a more explicit approach to recommendations for action from the findings - this is currently implied but not crystal clear. Even if the authors decide not to add an IR approach, I think more information in terms of dissemination with the aim of strengthening service provision and influencing local policy is important.

Response: We acknowledge the recommendation made by the reviewer. The main objective of this situation analysis protocol is to assess the current situation of SRH of Rohingya women and adolescent girls (aged 12-59 years) residing in the refugee camps, understand the community perspectives and facility readiness to provide different SRH services, related gaps and challenges. This situation analysis will inform World Health Organization (WHO) to design and deliver integrated comprehensive SRH services to meet the immediate SRH needs of these extremely vulnerable Rohingya women and adolescent girls. In the next phase, an implementation research will be conducted while the intervention is designed and being implemented. This protocol is expected to serve as a reference for future research in terms of methodological innovations, data collection instruments, insights and guide on how to conduct research on SRHR of this particular population and in similar humanitarian settings.

The additional recommendation by reviewer on Dissemination plan has been incorporated on new pages – 12 & 13.

2. The study is described as using concurrent mixed methods. Whilst this is fine as an approach, and the purpose is clearly complementarity and triangulation of results, the data analysis and interpretation section has no information about how the findings from the data collection methods will be integrated and interpreted jointly. So, either describe it a multi methods study design within an IR framework, or if mixed methods, need to describe data integration as part of analysis.

Response: We accept the comment from reviewer on expanding the data analysis section. We have now incorporated a subsection 'Data Triangulation' in the Methods section on new page 12.

3. I found it a bit confusing jumping between the three data collection methods in each of the sampling, data collection and data analysis sections.

I wondered whether it would be clearer to introduce the three data collection methods, and then describe each in total in turn.

That is - household survey - sampling strategy, data collection tool, data collection and data analysis.

Qualitative interviews - sampling strategy, data collection, data analysis,

Facility survey - etc etc.

Response: We acknowledge the point raised by the reviewer and revision on the Methods section has been done accordingly on pages 6–11.

4. In terms of qualitative analysis - it sounds like the authors are describing a kind of abductive analysis as described by Tavory and Timmermans (2014) - ie. initially deductive coding using a priori codes, but then allowing new codes to arise and adding them.

Response: Yes, as rightly identified by the reviewer, the study protocol describes the qualitative analysis using deductive coding approach using a priori codes but will also keep emergent codes to evolve from data (i.e. reading transcripts data).

5. Small thing, but in the ethics section, should be assent rather than ascent for adolescent girls.

Response: The correction on the identified section (page – 12) has been made.

Reviewer: 2

Reviewer Name: Brynne Gilmore

Institution and Country: Centre for Global Health, Trinity College Dublin

Ireland

Major:

1. The limitation listed in Abstract is not a 'methodological' limitation per se. I would rephrase to just, "a potential limitation foreseen within this study is the unwillingness...".

Response: Thanks for the comment; rephrased as suggested by the reviewer.

2. It would be helpful to explain the difference between comprehensive and basic package of SRH in the literature/background.

Response: Thanks, the explanations are added on pages – 4 & 5.

3. In the abstract, you note that the focus will be on Family Planning and safe abortion services, but this is not discussed anywhere else within the manuscript. Please note how this focus will be seen within your methodology, or revise accordingly.

Response: Thanks for notifying. The necessary revision has been made in the beginning of Page – 5, in the Background section, not in the Abstract.

4. There is very limited information on the study sites. Please expand to ensure the readers have a more accurate picture of the context.

Response: Thanks. An expansion on the current context of the study sites have been added on Page – 6.

5. Please review for grammar. While the article is well written, at times there are awkward sentences or those that could benefit from addition, or removal, of some punctuation. I have not identified all the sentences within this report.

Response: Thanks for the note. The necessary copyediting has been done as identified by reviewer.

6. Please explain where the 22% that gave birth in health facilities factored into your sample size calculation (I think you're saying this was used as prevalence rate, but a little unclear), and why you used this as your rate? I thought comprehensive SHR services would go beyond facility? Also, if your primary focus (as stated in intro) is family planning and abortion, wouldn't these estimates be better to use for sample size?

Response: Thanks for the comment. Firstly, primary focus of the study is not only on family planning and abortion, rather to capture overall scenario of SRH situation and needs that includes pregnancy, delivery care, postnatal care, family planning, abortion and menstruation within the specified study population (Rohingya women and adolescent girls aged 12-59 years). However, while designing the study, prevalence rates on SRH outcomes and utilizations were explored, no prevalence on family planning, contraception use, or abortion were found. Only delivery rate at health facilities were available in the existing literature body, thus, this was factored into sample size calculation. This point has been added to the Strengths and Limitations section in the beginning of Page – 3.

7. Sampling – several issues with this: 1) Please provide more clarity on methods. You are doing multi-stage sampling with both cluster and simple random sampling it appears?; 2) You note (line 6, pg. 9) 'proportionality', but there is no indication that you are doing proportionate sampling. If not, justify why you are not intending to do proportionate

sampling as I presume both the camp and community sizes vary; 3) Facility sampling is very confusing –are you doing one of each category (5 categories), in each of your camps? Or in total? I am unclear how you are getting the numbers at the end of this section; 4) I think that all sampling sections section can be simplified (it is very long, and also includes a diagram) and clarified a fair bit. It is too wordy and somewhat confusing.

Response: Regarding points 1 & 2, this study does not include any component of cluster sampling. It will employ a three-stage simple random sampling method for household survey. We are also unable to conduct a proportionate sampling depending on population size of each camp. Due to constant movement of refugees from camp-to-camp and constraints related to time, resources and available logistics for the research project, it was not possible to proportionately sample the population at all study camps. The term “proportionality” used in the identified paragraph (on new page – 7) refers to proportional distribution of respondents in adolescent (12 – 17 years) and women (18 – 59 years) groups in every camp.

Facility assessment will be done in the similar camps as the household survey. Sampling of health facilities will be done depending on availability in each camp as per the five categories mentioned. We attempted to assess at least one facility from each category in every camp, however, due to unavailability of health facilities, it was not possible in some cases. This has been explained on new pages 10 & 11 in sample size and sampling techniques sub-section under Facility Assessment section with an additional table in the main text (Table -3) on page – 11.

8. Unclear why including an adolescent from the same Household, might need more explanation and/or note the potential limitations of this (see point below on this specifically)

Response: We acknowledge the concern raised by the reviewer. However, considering the religious and cultural conservative nature of the Rohingya community, including adolescent girls from one household without interviewing the mother/other adult woman was not feasible. During pretesting it was observed that adolescents’ mothers were suspicious to let the adolescent girl speak to the interviewer alone for the interview period. Most of the times, they were accompanying the adolescents which hampered privacy and adolescents were uncomfortable to respond to SRH specific queries. On the other hand, while we attempted to pretest by including adult woman and adolescent girls from the same household, accessibility and privacy to interview the adolescents became very easy as the mother was not anymore suspicious rather thought speaking to the interviewers would be helpful for the adolescents.

9. Line 16, pg. 9 – married women, this was not noted in your participation criteria that they had to be married (see note below on study populations).

Response: The correction has been made accordingly on new page – 7.

10. Please include your intended study dates.

Response: Study dates has been included on new page – 6.

11. Ethics – I am unsure if this study poses no more than minimal risk. I can imagine SRH topics (especially FP and abortion), can be quite risky to discuss and may in fact put people at risk if

any information was to be released. How will you make sure people are aware of this risk, and what measures will you put in place to protect them? How will you collect consent for persons with low literacy?

Response: BRAC James P Grant School of Public Health has a Centre of Excellence for Gender, Sexual and Reproductive Health and Rights (CGSRHR) set up since 2008 that trains researchers to be skillful in conducting research related to SRH issues and how to minimize any associated risks. Our research assistants will be trained on the sensitivity of SRH specific topics such as family planning and abortion. They will also be coached on the religious and cultural sensitivity of the Rohingya community, and how to maintain privacy and confidentiality along with anonymity of the respondent. Any question/query that raises risk related to privacy of the respondent (specifically related to family planning and abortion) has been revised during pretesting, so that the respondents are posed with no more than minimal risk. Regarding data protection, controlled access folders and locked storage with only authorized access (Principal Investigator, other study investigators, and IRB members of BRAC JPGSPH) will be utilized.

Verbal consents will be sought from respondents with low literacy. The necessary amendments on the Ethics section has been made on new page – 12.

12. Please clarify ethics – have you sought (pg. 12 line 9) or have you been granted (pg. 14).

Please provide proof of ethics via certificate (or at least approval number).

Response: This study has been granted ethical approval from Institutional Review Board (IRB) of BRAC James P Grant School of Public Health (2018-017-IR). The necessary changes have been done on the content on new page –12.

13. The synthesis/situational assessment bit of the study is lacking. What is your plan for synthesising your data to provide an overall 'situation analysis'. Do you have any plans for looking at the specific camps data (i.e. are there relations between 'stronger' facilities and the survey respondents), etc.

Response: Analyzing camp-specific data is not planned in this study protocol. We plan to capture an overall scenario of SRH needs and demands in the Rohingya refugee camps. This study sample is thus planned to be representative to describe an overall picture using both qualitative and quantitative methods. Initially, both qualitative interviews, discussions and quantitative survey, assessment will be analyzed individually and later, all data collected using various methods will be triangulated as explained on new page – 12.

14. Under data collection methods - please indicate the topics that will be included within the

Household survey, as this section has very little about the tool itself, only on the collection

method. If you have any of your tools developed (facility assessments or quantitative) it would be good to include as supplementary file.

Response: Necessary information related to Household survey tool has been added to the content on new page – 6. Quantitative tools (both household survey and facility assessment) will be uploaded as supplementary file during submission of revised manuscript.

15. Age-sets: as noted below under study considerations, I am cautious that considering all women age 18-59 as one homogenous group (despite there being a 41 years age range), does not go towards person-centered SRH service design. Please include how you will ensure you are collecting data that can contribute to designing such SHR comprehensive service packages.

Response: The authors acknowledge the comment made by the reviewer. In order to collect person-centered data, designing the study with segregated age-sets, we have to increase our sample size which is not feasible logistically considering time and resource constrain. However, while analyzing our findings will showcase information according to different segregated age groups for both adolescents and women.

Minor:

1. Line 7-8 in the abstract – what lessons learned are you trying to capture? From the minimum initial package, for scaling up to the comprehensive? This sentence could use some rewording. Also, what current dimensions? Of the package, of the context...? Last sentence of abstract – first line is awkward and needs grammar check. Also, is the situation analysis providing information relevant to similar low-and middle-income contexts, or to similar humanitarian crisis?

Response: Thanks. Necessary changes are made on the section identified in the Abstract on new page – 2. The dimension is of the context, on their SRH needs and challenges related to comprehensive SRH package implementation. Last sentence has been revised. The situation analysis will provide information relevant to similar humanitarian crisis context as rightly noted by the reviewer.

2. More details on your methods would be helpful in the abstract. What kind of community based survey? What is the purpose of it? Would help to note within abstract. What types of interviews and discussions? (note KIIs, IDIs and FGDs as per table 2).

Response: Due to word limit few corrections could be made on the survey and interview related points in the Abstract on new page – 2.

3. Who are the stakeholders involved in the survey and the discussions (you note community people, but you have indicated numerous stakeholder groups within the body).

Response: Due to word limit relevant stakeholders are mentioned in the Abstract which has been elaborated in the methods section later within the manuscript body on page – 9.

4. Point two “this study employs a mixed-method...” under Strengths and Limitations is somewhat awkward, please review grammar.

Response: Reviewed and revised.

5. Abstract – service package, should this be service packages? Again in 7th line.

Response: Revised.

6. SD Goals to SD Goal, line 6?

Response: Revised.

7. Please review your referencing style. Typically, the full stop goes after the citation (i.e.

humanitarian response [7].”

Response: We followed the referencing style instructed by BMJ Open on its “References” section in “Formatting your paper.” Link: <https://authors.bmj.com/writing-and-formatting/formatting-your-paper/>

8. Line 16-18 – would they be classified as refugees then? According to your definition, no.

Maybe change to ..’of which 4.4 million are newly displaced persons’.

Response: Thanks. Revised as suggested by the reviewer in the Introduction section on new page – 4.

9. Line 20 – group to groups.

Response: Revised.

10. Pg. 7, line 19: is this to mean that your inclusion criteria is that they, a) must be forcedly displaced from Rakhine specifically, b) must not have left Myanmar before August 25, 2017.

What about women who have entered/experienced menopause, will they be included?

What about the inclusion of men (pg. 7 line 43)? Do they have to have a partner/spouse within this age group? What is their inclusion criteria?

Response: Any women or adolescent girls aged 12-59 years who have entered Bangladesh after August 25, 2017 will be included as study respondents. As refugees that entered Bangladesh before August 25, 2017 has been living in the country for quite a period and did not enter with the refugee influx after the violence on the date, will not be included as study sample. Women with menopause will also be included to understand their SRH needs and demands.

Rohingya adult males will be included as study respondents. Necessary change has been made on last row of new Table – 2, page – 9.

11. Pg. 8, line 15. Is that the total number all over, or just in your study sites.

Response: That is the total number all over, the study population, not only in the study sites.

12. Pg. 8, under Stage I: Would be better to report the inclusion/exclusion of camps first, and then the selection of the camps within the 'new makeshift camps'. Currently this is confusing, and makes it seem like you select 11, and then you select more from the new ones.

Response: Thanks for the comment. The changes have been made on new page – 7.

13. Table 2 – clarify that you mean 3 FGDs, not 3 participants (like I believe the rest of your columns indicate)

Response: Thanks. Changes have been made on last row of new Table – 2, new page – 9.

14. Line 26 pg. 9 – unsure what is meant by 'depending on geographic location...' are you trying to get only remote/difficult places, do you want a balance? Will the KIIs be from these 3 sites, or are KIIs not at this level?

Response: Qualitative data is planned to be collected from geographically remote/difficult places as well as from nearer camps in the Cox's Bazar district. KIIs with community influential will be from these 3 camps as well. However, KIIs with local and international NGO programme leads, managers, SRH focal points, influential workers, government high officials, and programme managers will not be confined to these 3 camp sites.

15. Pg. 10, line 15 – what do you mean 'based on proximity random selection will be conducted'?

Response: Revision has been done on new pages – 10 & 11 under "Sample size and sampling techniques" of Facility Assessment.

16. Pg. 10, line 17 – you list only one secondary facility, but note there are two.

Response: Along with 2 Government secondary health facilities, there are other secondary health facilities serving the study population, either owned or managed by national or international NGOs/private organizations.

17. Figure 1 (appendix) note that you will collect from 30 and 10 women and adolescents, respectively. This is different than in the body of the paper. Please review diagram.

Response: Thanks. The diagram has been revised accordingly.

18. Pg. 11, line 44 “Group discussions... tested using data displays...” please explain this?

Response: Thanks for the comment. On new page – 10, this section has been replaced with “Emerging themes and patterns in the data will be tested using data displays that allow more systematic analysis of the qualitative data.” Hope this would explain.

19. It could be important to include in your dissemination plans how you intend these findings to influence SRH programming/packages within Cox’s Bazar. Essentially – revisit why you are doing the situational analysis in the first place.

Response: Thanks. The Dissemination section has been updated as per suggestion by the reviewer on new pages – 12 & 13.

Study considerations:

- Consider your population groups – I imagine within Rohingya some married women and unmarried women might have different needs, and/or at least need to discuss these in different groups (same goes for men). Might therefore be good to distinguish between either ‘adolescent’ married or unmarried, and non-adolescent married and unmarried, within your interviews and IDIs (also relevant for pg. 7). This should also be considered for your FGDs with men – I would strongly consider conducting different ones (married, unmarried, adolescent, older etc). It could also be beneficial to try to complement your male FGDs with a couple IDIs with men, as often sensitive topics like family planning and abortion might be difficult to discuss in more open forms.

Response: Thanks for the comment. However, qualitative component of this research aims to explore deeper insights of SRH needs and demands of the Rohingya adolescent and women (aged 12 -59 years). The interview approach thus would be more exploratory towards the cultural and religious context of this population in addition to their other barriers and challenges in accessing SRH services in such humanitarian context. Age group-specific or married/unmarried group-specific data collection was not considered while designing the study due to the reported violence this population group has experienced while in Myanmar as well as in the camps and related variety of dynamics in their marital status reported in anecdotal data. For example, married women left Myanmar without their husbands, adolescents getting married while in camps etc. Thus, the investigators aimed to understand the

overall scenario of SRH specific needs and demands in this group rather than dividing them into different population groups.

Due to the objective of exploring men's perception and roles in SRH care-seeking of adolescent girls and women in the Rohingya community, focus group discussions (FGDs) have been included in the study. As we do not aim to explore individual level information from Rohingya males, conducting IDIs with them have not been considered.

- You may also need to do more than 3 FGDs with men. I would not be surprised if (at least within married couples), men can heavily influence SRH access and demand. However, this goes to a point below where I suggest designing/conducting qualitative after you have analysed surveys.

Response: The study will be using a concurrent mixed methods design. Conducting sequential data collection considering the study setting and population context (collecting qualitative data after analyzing survey data) would be tough and not feasible given time and resource limitation.

We acknowledge the fact that more than 3 FGDs could be needed. However, we shall collect qualitative data until data saturation is achieved, as identified on new page – 9 under Table – 2.

- As noted above, please consider not using adolescents from same households as nonadolescent.

First, they will know the types of questions you asked each other, which could entrench on ethical issues. Second, both respondents would likely have similar physical access to services (though obviously other barriers), which might skew some of your results.

Response: We acknowledge the concern raised by the reviewer. However, as mentioned earlier, considering the religious and cultural conservative nature of the Rohingya community, including adolescent girls from one household without interviewing the mother/other adult woman was not feasible. Mothers get suspicious to let the adolescent girl speak to the interviewer alone. Most of the times, they accompanied the adolescents that hampered privacy.

In regards to knowing the types of questions asked, the researchers plan to collect data simultaneously from the adult woman and adolescent girl from the same household with 2 research assistants. So that the ethical concern of knowing question earlier can be minimized.

Regarding, similar physical access to services, the available literature and pretesting of this study has shown adolescent girls in Rohingya refugee camps face different set of barriers compared to their adult counterparts.

- It might be helpful to analyse your quantitative data first, and then decide on some of your qualitative work, especially in terms of whom to interview and the study sites. You might find that different communities have different survey responses, depending on their access etc, so to only do 3 sites, when there could be variation in responses across say, 5 sites, would be limiting and not give you full picture. Further, Qualitative if you design your guidelines (and thus analysis

framework) after the quantitative has been analysed you could get more rich and purposeful data if you base some of the discussions on what was arising from the survey (i.e. can really focus on the main barriers are reported in survey, and why/how these arise, what can be done etc). This could give you the chance to do a more in-depth exploration of some of the key issues, and also serve to triangulate your previous findings. If this is something you would consider, then need to adjust some of the body to reflect this.

Response: Thanks for the suggestion. However, the tools (whether qualitative or quantitative) have been designed and finalized after several pretesting on the refugee camps. Designing a sequential method as suggested by the reviewer is not feasible given time and funding constrain, as already mentioned earlier. Same applied to the site selection for qualitative interviews and discussions. The study sites for qualitative interviews will be selected based on the criteria mentioned in the new page – 9 in Sample size and sampling techniques sub-section under Qualitative interviews and group discussions section. The authors expect that capturing qualitative information from these sites would help in getting an overall understanding of the community culture, barriers and challenges alongside the physical variations in the camp settings.

- Might be appropriate to stratify your analysis even further- after doing 18-59, could also do like 18-40? Or another age group. With this over 40-year age difference between 18-59, needs etc. are very likely to be different. Person-centred SHR services would need to be responsive to each age-set needs, and not treat like one homogenous group. I think you should also plan to do your interviews with various age-ranges.

Response: The authors plan to incorporate stratifying analysis of collected data in different age groups as identified by the reviewer to better identify their needs and recommend designing a responsive person-centered SRH service package.

VERSION 2 – REVIEW

REVIEWER	Brynne Gilmore Centre for Global Health, Trinity College Dublin, Ireland
REVIEW RETURNED	26-Mar-2019

GENERAL COMMENTS	Thanks you again for the opportunity to review this manuscript detailing important research to be undertaken. The authors have either incorporated my previous comments or explained why they were not appropriate to my satisfaction. Good luck with the rest of the process!
---